# Prevalence and Characteristics of Metabolic Hyperferritinemia in a Population-Based Central-European Cohort

**DOI:** 10.3390/biomedicines12010207

**Published:** 2024-01-17

**Authors:** Sophie Gensluckner, Bernhard Wernly, Florian Koutny, Georg Strebinger, Stephan Zandanell, Lars Stechemesser, Bernhard Paulweber, Bernhard Iglseder, Eugen Trinka, Vanessa Frey, Patrick Langthaler, Georg Semmler, Luca Valenti, Elena Corradini, Christian Datz, Elmar Aigner

**Affiliations:** 1Department of Internal Medicine I, Paracelsus Medical University, Müllner Hauptstrasse 48, 5020 Salzburg, Austria; s.gensluckner@salk.at (S.G.);; 2Obesity Research Unit, Paracelsus Medical University, Müllner Hauptstrasse 48, 5020 Salzburg, Austria; 3Department of Medicine, General Hospital Oberndorf, Teaching Hospital of the Paracelsus Medical University, Paracelsusstraße 37, 5110 Oberndorf, Austria; 4Department of Internal Medicine 2, Gastroenterology and Hepatology and Rheumatology, University Hospital of St. Pölten, Karl Landsteiner University of Health Sciences, Dunant-Platz 1, Kremser Landstraße 40, 3100 St. Pölten, Austria; 5Department of Geriatric Medicine, Christian Doppler University Hospital, Paracelsus Medical University, Ignaz-Harrer-Straße 79, 5020 Salzburg, Austria; 6Department of Neurology, Christian Doppler University Hospital, Paracelsus Medical University and Centre for Cognitive Neuroscience, Affiliated Member of the European Reference Network EpiCARE, Ignaz-Harrer-Straße 79, 5020 Salzburg, Austria; 7Vienna Hepatic Hemodynamic Lab, Division of Gastroenterology and Hepatology, Department of Internal Medicine III, Medical University of Vienna, Währinger Gürtel 18-20, 1090 Vienna, Austria; 8Department of Pathophysiology and Transplantation, Università degli Studi di Milano, Via Francesco Forza 35, 20122 Milan, Italy; luca.valenti@unimi.it; 9Precision Medicine, Biological Resource Center Unit, Fondazione IRCCS Ca’ Granda Ospedale Maggiore Policlinico, Via Francesco Sforza, 35, 20122 Milan, Italy; 10Department of Medical and Surgical Sciences, Università degli Studi di Modena e Reggio Emilia, Via del Pozzo 71, 41124 Modena, Italy; elena.corradini75@unimore.it; 11Internal Medicine and Centre for Hemochromatosis and Hereditary Liver Diseases, Azienda Ospedaliero-Universitaria di Modena Policlinico, 41124 Modena, Italy

**Keywords:** metabolic hyperferritinemia, fatty liver disease, MASLD, insulin resistance, metabolic syndrome, serum ferritin

## Abstract

Background: Hyperferritinemia (HF) is a common finding and can be considered as metabolic HF (MHF) in combination with metabolic diseases. The definition of MHF was heterogenous until a consensus statement was published recently. Our aim was to apply the definition of MHF to provide data on the prevalence and characteristics of MHF in a Central-European cohort. Methods: This study was a retrospective analysis of the Paracelsus 10,000 study, a population-based cohort study from the region of Salzburg, Austria. We included 8408 participants, aged 40–77. Participants with HF were divided into three categories according to their level of HF and evaluated for metabolic co-morbidities defined by the proposed criteria for MHF. Results: HF was present in 13% (n = 1111) with a clear male preponderance (n = 771, 69% of HF). Within the HF group, 81% (n = 901) of subjects fulfilled the metabolic criteria and were defined as MHF, of which 75% (n = 674) were characterized by a major criterion. In the remaining HF cohort, 52% (n = 227 of 437) of subjects were classified as MHF after application of the minor criteria. Conclusion: HF is a common finding in the general middle-aged population and the majority of cases are classified as MHF. The new classification provides useful criteria for defining MHF.

## 1. Introduction

Ferritin is a multimeric protein complex with essential functions in iron homeostasis and storage. Although only a small amount is found in serum, the measurement of serum ferritin (SF) is clinically useful as an indicator of the total body iron content [1]. Hyperferritinemia (HF) is defined as elevated SF concentrations and is present in about 5–25% of the general population [2]. Several different conditions may lead to HF and are classified as primary, i.e., inherited causes based on genetic variations, or secondary, i.e., acquired causes due to hematological, inflammatory or metabolic diseases [3]. 

Due to the rapidly increasing prevalence of obesity, type 2 diabetes mellitus (T2DM) and metabolic-associated steatotic liver disease (MASLD) worldwide [4,5], persistent HF is most commonly associated with these metabolic diseases. The elevation of ferritin in patients affected by these conditions is considered a consequence of combined genetic, alimentary and inflammatory stimuli on iron fluxes. Different factors contributing to the pathogenesis, as well as associations and relations of HF with components of these disease mechanisms, have been described [6,7,8,9,10,11]. Among metabolic-associated forms of HF, an additional increase in iron stores within the liver has been referred to as dysmetabolic iron-overload syndrome [12]. Several effects of iron overload on chronic metabolic conditions and organ damage have been reported, including a higher incidence of T2DM and insulin resistance and there are conflicting data on liver disease severity and cardiovascular events [13].

Although associations between HF and various metabolic conditions are well known, they have not been conclusively defined. There have neither been uniform cut-off values for a consistent grading of HF nor have the underlying metabolic conditions for a standardized diagnosis been defined. To overcome those gaps, Valenti et al. recently published a consensus statement on the definition and classification of metabolic hyperferritinemia (MHF), which may provide a broadly accepted working definition of MHF [14]. 

Based on this novel definition, we evaluated the data of the Paracelsus 10,000 (P10) study from the region of Salzburg, Austria. The aim was to generate information on the prevalence of MHF in a Central-European cohort and to provide characteristics of associated metabolic co-morbidities in order to evaluate the proposed classification.

## 2. Materials and Methods

### 2.1. Study Population

Data was used from the P10 study, which is a population-based observational cohort study of the city and surrounding region of Salzburg, Austria. The objective of the P10 study is to investigate the general health of the population, with a particular focus on metabolic, cardiovascular and cerebrovascular diseases. The recruitment phase ended in 2020, and during follow-up, clinical events were recorded. Participants were invited after random selection from the local population registry, and 10,044 participants aged 40–77 years were recruited. All participants signed an informed consent form. A detailed description of the study was provided recently [15]. 

For this analysis, the baseline data set was used. The study team had access to all data and ensured its protection and integrity. The P10 study protocol was approved by the local ethics committee (415-E/1521/3-2012), and all study-related procedures were carried out in accordance with the principles of the Declaration of Helsinki. 

### 2.2. Inclusion and Exclusion Criteria

Analyses were performed at three different levels of patient selection: (i) the overall cohort, (ii) the study cohort and (iii) the affirmation cohort.

(i)Overall cohort: A baseline serum ferritin concentration was available in n = 9915 participants, comprising our overall cohort and representing an unselected population level.(ii)Study cohort: For the study cohort, we applied the consensus exclusion criteria, which were (a) transferrin saturation of >50% (n = 776), (b) anemia defined as blood hemoglobin < 12.5 g/dL in women and <13.0 g/dL in men (n = 396), (c) advanced chronic kidney disease defined by an estimated glomerular filtration rate (eGFR) of <30 mL/min (n = 10, with n = 1 with an eGFR < 15 mL/min) or (d) self-reported daily alcohol intake of >40 g in women and >60 g in men (n = 416). Some individuals held more than one exclusion criterion. Thus, after exclusion of a total of n = 1507 participants, the study cohort consisted of n = 8408 subjects. (iii)Affirmation cohort: As inaccuracies may arise from missing values with regard to the metabolic classification, we aimed to validate our findings in a cohort of subjects, in which a full metabolic characterization due to entirely complete data sets of the proposed metabolic parameters were available. This cohort comprised n = 6424 participants. Thus, any misclassification due to potentially incomplete metabolic data was avoided.

HF was defined as SF > 200 ng/mL in females and SF > 300 ng/mL in males. For further analysis, participants with HF were grouped into the three grades of HF according to the consensus: (1)HF grade 1 with SF < 550 ng/mL (females 200–549 ng/mL, males 300–549 ng/mL)(2)HF grade 2 with SF of 550–1000 ng/mL(3)HF grade 3 with SF > 1000 ng/mL.

Figure 1 shows a chart of the selection of the study cohort.

### 2.3. Definition of Metabolic Hyperferritinemia

Hyperferritinemia was defined and considered as metabolic according to the recent consensus. Table 1 shows a summary of the diagnostic criteria, and the parameters used in this study are highlighted. In brief, the criteria were summarized as presence of a major feature of metabolic dysfunction, like evidence of fatty liver and/or T2DM and/or obesity, or the presence of at least two minor features of altered metabolism, like overweight or increased abdominal circumference, elevated triglycerides, low HDL cholesterol, increased fasting glucose, arterial hypertension or elevated HOMA-Index.

For this analysis, we first applied the major criteria to classify those with manifest metabolic disease. Liver steatosis was defined with a fatty liver index (FLI) > 60 because no other techniques were available. The FLI is a non-invasive, validated biochemical assessment of liver steatosis, using triglycerides, gamma-glutamyltransferase, waist circumference and BMI (body mass index) [16,17]. All other parameters were available from the baseline evaluation of the P10 study. T2DM was defined according to national guidelines with either HbA1c ≥ 6.5% or an elevated fasting glucose of ≥126 mg/dL or a 2 h post glucose load glycemia of ≥200 mg/dL during oral glucose tolerance test or use of antidiabetic medication [18]. Obesity was defined as BMI ≥ 30 kg/m^2^ [19]. After application of the major criteria, the minor criteria were applied to the remaining subjects. A list and definition of the minor criteria is provided in Table 1. 

### 2.4. Statistical Analysis

Continuous data are given as median ± interquartile range (IQR) and compared using Mann’s Whitney U-Test or mean ± standard deviation (SD) and compared using Student’s *t*-Test accordingly. Categorical data are given as numbers (percentage) and compared using the Chi-square test. All tests were two-sided, and a *p*-value of <0.05 was considered statistically significant. Stata/IC 17 was used for all statistical analyses.

## 3. Results

### 3.1. Characteristics of the Overall Cohort

A baseline SF level was available in n = 9915 subjects. Within this overall cohort, 86% of (n = 8527) participants had normal SF levels (n = 3845/45% male, n = 4682/55% female), while 14% (n = 1388) of participants had elevated SF, of which 66% (n = 917) were male. 

Among subjects with HF, 88% (n = 1225) were classified as HF grade 1, 11% (n = 146) as HF grade 2 and 1% (n = 17) as HF grade 3, respectively. Subjects within the non-HF group were younger (mean age 55 ± 8) than the subjects within the HF group (mean age 58 ± 8 for HF 1, HF 2 and HF 3, respectively). Generally, MHF defining criteria (major and minor) were found in 55% (n = 4692) of subjects of the non-HF group, compared to 78% (n = 960), 92% (n = 134) and 94% (n = 16) of subjects within the HF groups grade 1, grade 2 and grade 3, respectively. Table 2 shows the baseline characteristics, and Table 3 shows the prevalence of major and minor metabolic characteristics of the overall cohort. Classification by minor criteria was only applied to subjects who had not been classified by a major criterion.

### 3.2. Characteristics of the Study Cohort Using the MHF Defining Criteria

According to the abovementioned inclusion and exclusion criteria, n = 8408 participants were eligible for analysis regarding MHF, of which n = 1111 (13%) fulfilled the criteria of HF. Within the subjects with HF, the majority, 90% (n = 997), were classified as HF grade 1, in comparison to 9% (n = 107) with HF grade 2 and 0.5% (n = 7) with HF grade 3. The mean age of subjects with and without HF was similar, at 56 ± 8 years in the non-HF-group versus 58 ± 8 years, 57 ± 8 years and 56 ± 4 years in the HF groups grade 1, grade 2 and grade 3, respectively. There was a clear male preponderance within the HF group, at 69% (n = 771) in comparison to 44% (n = 3243) men in the non-HF group. Table 4 shows the baseline and anthropometric characteristics of our study cohort.

The results with regard to application of the metabolic criteria on the HF groups are shown in Table 5. The differences between the HF and non-HF group were statistically significant, except for the minor criterion low HDL cholesterol. In general, subjects with HF fulfilled the metabolic criteria more often, at 81% (n = 901 of 1111), in comparison to the non-HF group at 56% (n = 4084 of 7297). Fulfilled metabolic criteria were found in 59% (n = 4984) of the general study cohort, but prevalence increased along the grades of HF with fulfilled metabolic criteria in 82% (n = 795) of the population with HF grade 1, 95% (n = 99) with HF grade 2 and 100% (n = 7) with HF grade 3, respectively. 

Major metabolic characteristics were more frequent in the HF group, at 61% (n = 673) compared to 32% (n = 2300) in the non-HF group, and prevalence again increased among the grades of HF. In the remaining participants after application of the major criteria (n = 5435), any combination of at least two minor metabolic criteria was found in 52% (n = 227 of 437) of the remaining HF group and 36% (n = 1784 of 4997) of the remaining non-HF group. 

In summary, 81% (n = 901 of 1111) of subjects within the HF group fulfilled the diagnostic metabolic criteria and may therefore be classified as MHF, of which the majority, 75% (n = 673 of 901), were characterized by the presence of a major criterion. A detailed listing of general prevalence and proportions of the major criteria within the study cohort and the minor criteria within the remaining cohort is shown in Table 5. 

### 3.3. Characteristics of the Affirmation Cohort

For validation of our findings due to missing values, we additionally analyzed a smaller subset of n = 6424 subjects without any missing values with regard to the data required for metabolic characterization. The results did not differ from those of our study cohort. A description of this affirmation cohort is provided in the Appendix A.

## 4. Discussion

Our data confirm HF as a frequent laboratory finding at the population level and metabolic alterations as its leading underlying cause, with increasing prevalence among the severity of HF. Although similar observations have been published in the past, no uniform definition for MHF has been provided so far. Dysmetabolic hyperferritinemia (DHF) was first described when HF with normal transferrin saturation was linked to metabolic disorders [20] and has further been referred to as insulin resistance-associated hepatic iron overload with additional positive histological iron staining [21]. Later, the definition of dysmetabolic iron overload (DIOS) was refined as HF with normal or mildly increased transferrin saturation, metabolic abnormalities and a specific amount of hepatic iron excess with liver biopsy or MR imaging [12]. Many reports followed describing the underlying pathophysiology, metabolic alterations [22,23,24,25,26,27,28], the impact of DHF on liver damage [29], or the controversial therapeutic effect of venesection [30]. However, so far, different definitions and criteria for MHF have been used, making comparisons difficult. Additionally, the necessity of iron quantification by biopsy or MRI has complicated diagnosis due to the limited availability of both methods. The proposed new definition is based on the consensus of involved researchers and promises to overcome those difficulties by suggesting uniform cut-off values and a clear listing and definition for underlying metabolic conditions to help standardize clinical diagnosis.

We therefore applied the novel definition to provide data on MHF from a population-based cohort. The major strength of our study is its population-based design, which provides findings without preselection and allows conclusions for a general Central-European population. In general, numbers and percentages for the prevalence of metabolic alterations were similar within the overall population and the study cohort. The prevalence of HF of 13% in our study cohort (and 14% in the overall cohort) is in line with data reported so far. The prevalence of severe HF with SF > 1000 mg/dL was low, with 17 out of 9915 subjects affected on a population level for the overall cohort and 7 out of 8408 for the study cohort, accounting for just 0.5% and 1% among subjects with HF, respectively. However, the whole group of MHF grade 3 in our study cohort was classified as MHF by a major metabolic feature, underlining the relevance of metabolic alterations in severe HF besides potential other acquired conditions. Furthermore, this grade may constitute an important subset for further studies regarding additional genetic predisposition of iron metabolism disorders and iron accumulation.

Of all subjects with HF, the vast majority, 81%, were classified as MHF. Therefore, our data confirm underlying metabolic conditions as the most common reason for HF, although this high prevalence does not reflect a cause–effect relation. In three out of four subjects with MHF, a major criterion was the defining feature. In our cohort, the most common major criterion was fatty liver disease, with an overall prevalence of 32%, which is in line with known prevalence worldwide [31]. Hence, these numbers also support that FLI serves as an adequate means to estimate the presence of fatty liver on the population level. Of all subjects with MASLD (n = 2669) in our cohort, 24% (n = 636) had HF, which again supports previous estimates of HF prevalence in MASLD [32]. We also observed a higher prevalence of MASLD of 57% in subjects with HF and a further increase in MASLD prevalence along with the severity of HF [29,33,34,35]. A relationship between intravenous iron therapy and worsening of hepatic fat was also shown in the setting of dialysis [36] and recently a relationship between liver iron and MASLD was proposed by Mendelian randomization analysis [37]. It will be particularly worthwhile to analyze these genetic interrelations at the population level. 

The numbers of the major criteria of obesity and T2DM also are in line with numbers reported previously for Europe [4,20,38]. Similarly to MASLD in subjects with HF, the prevalence of obesity and T2DM was significantly higher in the HF group, likely due to common underlying mechanisms like subclinical inflammation with changes in the hepcidin–ferroportin axis, abnormal endocrine function of adipose tissue and insulin resistance [6,39]. 

Of the remaining participants after application of the major criteria, one-half were additionally classified as MHF due to the minor criteria. The criteria of overweight, arterial hypertension, impaired fasting glucose and elevated HOMA, again, were significantly more frequent among participants with HF. Interestingly, the two least-common minor features, low HDL cholesterol and triglycerides, showed less or even no significant differences between groups. We assume an essential part of those with altered lipid metabolism were already classified with a major criterion like MASLD and hence these subjects were not counted again with these remaining minor criteria, resulting in lower numbers for respective minor categories within the HF group. And although metabolic alterations were also found in 56% of the non-HF group, we show a relevant association and increasing likelihood of HF in subjects with metabolic comorbidities at the population level.

There are also some limitations to our study. First, our findings are based on the results from one single baseline examination and were not re-evaluated after 3 months of lifestyle modification or resolution of a trigger, as suggested by the consensus. This might be particularly relevant for the influence of regular alcohol intake as we observed an increasing amount of daily alcohol consumption among the HF grades. Secondly, liver steatosis was only diagnosed non-invasively with FLI and it has to be considered that the same clinical and biochemical parameters like BMI or triglycerides are also part of the MHF defining criteria. Furthermore, a substantial number of approximately 15% of the overall cohort had to be excluded according to the suggested exclusion criteria. However, numbers and percentages for metabolic characteristics were similar in the overall and the study cohort. The high number of exclusions due to elevated transferrin saturation (n = 776) is noticeable. Even presuming a substantial part of those being caused by an increased amount of daily alcohol consumption (n = 416), the other half remains without explanation. We suggest that this exclusion criterion will need further study as it is likely that the majority of these subjects in fact also had MHF rather than hemochromatosis. We also had to handle a relevant number of missing values necessary for metabolic characterization. To provide information on MHF which was as accurate as possible at the population level, our study cohort included every subject with available baseline SF, except those who met any of the exclusion criteria. This was at the expense of missing values as a full metabolic characterization for every subject was available in n = 6424 of the population. However, the findings for MHF classification in this affirmation cohort did not differ from those of our study cohort, therefore supporting our results. Due to the descriptive scope of this study, we did not aim to perform multiple testing; a subgroup analysis between the non-HF group and each of the HF subgroups (HF 1, HF 2 + HF 3) is added in the Appendix A. In summary, our three-step approach demonstrated little variability in the rates for MHF classification, making the data a robust estimate of these criteria at the population level in a Central-European cohort.

## 5. Conclusions

In conclusion, we provide reliable data on the prevalence of HF in general and the characteristics of MHF in particular, highlighting the predominance of metabolic alterations as one underlying cause of HF. The consensus offers a standardized definition of MHF with harmonized criteria that are easy to apply. It is highly based on routine laboratory testing or easily available imaging methods and non-invasive techniques. Overall, the inclusive design with positive criteria for MHF is particularly appealing and makes the definition useful not just for research questions but also for clinical routines and might help in addressing unanswered questions about the impact of MHF on liver disease severity, cardiovascular disease or as a potential value of venesection.

## Figures and Tables

**Figure 1 biomedicines-12-00207-f001:**
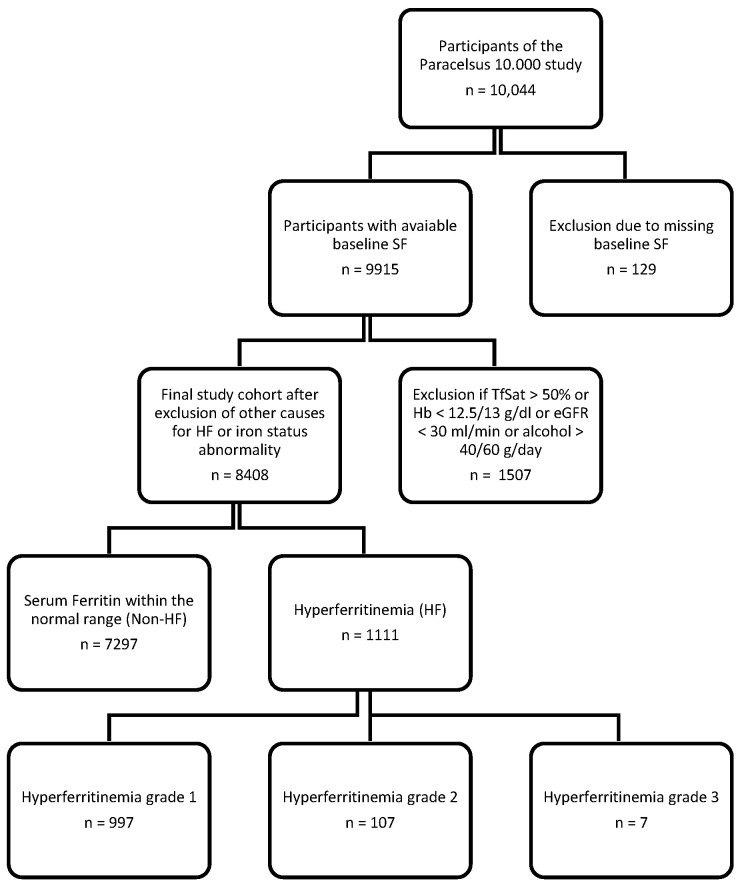
Flowchart with inclusion and exclusion criteria of the final study cohort. Abbreviations: HF hyperferritinemia, SF serum ferritin, TfSat transferrin saturation, eGFR estimated glomerular filtration rate, Hb hemoglobin.

**Table 1 biomedicines-12-00207-t001:** Diagnostic criteria for metabolic hyperferritinemia. Abbreviations: MRI magnetic resonance imaging; OGTT oral glucose tolerance test, BMI body mass index; HDL high-density lipoprotein; HOMA homeostasis model assessment.

Hyperferritinemia If Serum Ferritin Levels > 200 ng/mL in Females or >300 ng/mL in Males. Considered as Metabolic Hyperferritinemia If:	
Evidence of fatty liver	Liver biopsy. Imaging (Ultrasound, MRI, computed tomography). Continuous attenuation parameter on transient elastography. Non-invasive biomarker/scores (e.g., fatty liver index).
OR evidence of type 2 diabetes mellitus	HbA1c ≥ 6.5%. Fasting glucose ≥ 126 mg/dL. Two hour post glucose load glycemia of ≥200 mg/dL at OGTT. Use of antidiabetic medication.
OR evidence of obesity (BMI > 30 kg/m^2^)	BMI > 30 kg/m^2^_._
OR ≥ 2 of the following features of altered metabolism	Overweight (BMI ≥ 25 kg/m^2^) or increased waist circumference (>102/88 cm in M/F). Increased circulating triglycerides (150 mg/dL). Low HDL cholesterol (<45/55 mg/dL in M/F). Increased fasting glucose levels (>100 mg/dL). Arterial hypertension (>130/85 mmHg or use of anti-hypertensive agents). Evidence of fasting hyperinsulinemia or IR (locally validated, e.g., HOMA-IR index > 2.5).

**Table 2 biomedicines-12-00207-t002:** Characteristics for the overall cohort in total numbers and percentages (%) or median and interquartile range (IQR). Abbreviations: HF hyperferritinemia, BMI body mass index, Waist waist circumference, ALT alanine transaminase, AST aspartate transaminase, Gamma GT Gamma-glutamyltransferase, ALP alkaline phosphatase, HDL high-density lipoprotein, LDL low-density lipoprotein, TG triglycerides, Hgb blood hemoglobin, WBC white blood cells, PLT platelets. The asterisk (*) denotes significant difference between the non-HF and the HF group at *p* < 0.05.

	Non-HF n = 8527	HF All Grades (n = 1388)		
		HF 1 (n = 1225)	HF 2 (n = 146)	HF 3 (n = 17)
Age 40–49 years	25% (n = 2125)	All grades 17% (n = 234)		
		17% (n = 206)	18% (n = 26)	12% (n = 2)
Age 50–59 years	43% (n = 3684)	All grades 40% (n = 554)		
		40% (n = 494)	38% (n = 53)	41% (n = 7)
Age 60–69 years	28% (n = 2350)	All grades 37% (n = 523)		
		37% (n = 454)	42% (n = 62)	41% (n = 7)
Age ≥ 70 years	4% (n = 353)	All grades 6% (n = 77)		
		6% (n = 71)	3% (n = 5)	6% (n = 1)
Male *	45% (n = 3845)	All grades 70% (n = 971)		
		67% (n = 815)	95% (n = 139)	100% (n = 17)
Female *	55% (n = 4682)	All grades 30% (n = 417)		
		33% (n = 410)	5% (n = 7)	0% (n = 0)
BMI kg/m^2^ *	25 (23–29)	28 (25–31)	29 (26–32)	29 (27–32)
Waist cm *	92 (83–100)	99 (92–107)	102 (96–112)	104 (99–110)
Alcohol g/d *	7 (2–17)	13 (4–27)	15 (6–30)	38 (20–81)
ALT U/L *	21 (16–28)	28 (21–39)	38 (28–53)	55 (35–105)
AST U/L *	22 (19–27)	25 (21–31)	30 (24–39)	52 (33–141)
GammaGT U/L *	21 (15–32)	32 (22–51)	46 (29–73)	170 (58–402)
ALP U/L *	63 (53–75)	65 (56–77)	63 (51–75)	68 (62–98)
HDL mg/dL *	63 (52–75)	55 (46–68)	48 (42–55)	53 (44–70)
LDL mg/dL *	139 (115–164)	148 (124–171)	142 (116–174)	155 (121–170)
TG mg/dL *	94 (69–132)	118 (89–167)	160 (123–218)	197 (102–264)
Hgb g/dL *	14.2 (13.4–15.0)	14.9 (14.1–15.6)	15.3 (14.7–16.1)	15.2 (14.6–16.5)
WBC 10⁹/L *	5.8 (4.9–6.9)	6.1 (5.1–7.2)	5.9 (5.3–7.1)	6.4 (5.7–6.9)
PLT 10⁹/L *	247 (213–283)	237 (205–271)	210 (181–244)	198 (157–218)

**Table 3 biomedicines-12-00207-t003:** Summary of rates and proportions of major and minor metabolic characteristics of the overall cohort. The prevalence of the major characteristics is calculated for the overall cohort and the prevalence of the minor characteristics for the remaining cohort after application of the major characteristics. Within the two cohorts of major or minor criteria, an individual can fulfill more than one criterion but may only be classified once. Abbreviations: HF hyperferritinemia, FLI fatty liver index, BMI body mass index, T2DM type 2 diabetes mellitus, HOMA homeostasis model assessment, HDL high-density lipoprotein. The *p*-value indicates significant difference between the non-HF and the HF group at *p* < 0.05.

**Major Criteria within the Overall Cohort n = 9915**							
		**Overall Cohort** **n = 9915**	**HF All Grades n = 1388**			**Non-HF** **n = 8527**	***p*-Value**
			**HF 1** **n = 1225**	**HF 2** **n = 146**	**HF 3** **n = 17**		
**Major criteria**	FLI > 60	31% n = 3121	All grades 57% (n = 792)			27% (n = 2329)	**<0.001**
			54% (n = 661)	79% (n = 115)	94% (n = 16)		
	BMI > 30 kg/m^2^	19% n = 1923	All grades 31% (n = 433)			17% (n = 1490)	**<0.001**
			30% (n = 367)	40% (n = 59)	41% (n = 7)		
	T2DM	6% n = 593	All grades 10% (n = 140)			5% (n = 453)	**<0.001**
			9% (n = 109)	18% (n = 26)	29% (n = 5)		
**Minor Criteria within the Remaining Cohort n = 6436**							
		**Remaining cohort** **n = 6436**	**Remaining HF all grades n = 555**			**Remaining non-HF** **n = 5881**	***p*-value**
			**HF 1** **n = 525**	**HF 2** **n = 29**	**HF 3** **n = 1**		
**Minor criteria**	Overweight	43% n = 2758	All grades 57% (n = 315)			42% n = 2443	**<0.001**
			55% n = 303	2% n = 11	100% n = 1		
	Arterial hypertension	44% n = 2813	All grades 52% (n = 291)			43% n = 2522	**<0.001**
			49% n = 272	3% n = 19	0% n = 0		
	Elevated HOMA Index	13% n = 805	All grades 21% (n = 114)			12% n = 691	**<0.001**
			19% n = 108	1% n = 6	0% n = 0		
	Impaired fasting glucose	14% n = 876	All grades 23% (n = 126)			13% n = 750	**<0.001**
			22% n = 121	1% n = 5	0% n = 0		
	Low HDL cholesterol	11% n = 729	All grades 14% (n = 77)			11% n = 652	**0.048**
			13% n = 70	1% n = 7	0% n = 0		
	Elevated triglycerides	8% n = 544	All grades 11% (n = 60)			8% n = 484	**0.037**
			11% n = 59	1% n = 5	0% n = 0		

**Table 4 biomedicines-12-00207-t004:** Characteristics of the study cohort in total numbers and percentages (%) or median and interquartile range (IQR). Abbreviations: HF hyperferritinemia, BMI body mass index, Waist waist circumference, ALT alanine transaminase, AST aspartate transaminase, Gamma GT Gamma-glutamyltransferase, ALP alkaline phosphatase, HDL high-density lipoprotein, LDL low-density lipoprotein, TG triglycerides, Hgb blood hemoglobin, WBC white blood cells, PLT platelets. The asterisk (*) denotes significant difference between the non-HF and the HF group at *p* < 0.05.

	Non-HF n = 7297	HF All Grades (n = 1111)		
		HF 1 (n = 997)	HF 2 (n = 107)	HF 3 (n = 7)
Age 40–49 years	25% (n = 1789)	All grades 17% (n = 185)		
		17% (n = 166)	17% (n = 18)	14% (n = 1)
Age 50–59 years	43% (n = 3160)	All grades 41% (n = 453)		
		40% (n = 402)	44% (n = 47)	57% (n = 4)
Age 60–69 years	28% (n = 2032)	All grades 37% (n = 408)		
		37% (n = 369)	35% (n = 37)	29% (n = 2)
Age ≥ 70 years	4% (n = 307)	All grades 6% (n = 65)		
		6% (n = 60)	5% (n = 5)	0% (n = 0)
Male *	44% (n = 3243)	All grades 69% (n = 771)		
		66% (n = 661)	96% (n = 103)	100% (n = 7)
Female *	56% (n = 4054)	All grades 31% (n = 340)		
		34% (n = 336)	4% (n = 4)	0% (n = 0)
BMI kg/m^2^ *	26 (23–29)	28 (25–31)	30 (26–32)	31 (27–34)
Waist cm *	92 (83–101)	100 (93–107)	102 (96–112)	105 (102–112)
Alcohol g/d *	6 (2–15)	11 (4–21)	13 (4–26)	31 (15–38)
ALT U/L *	21 (16–28)	28 (21–39)	38 (28–58)	35 (25–92)
AST U/L *	22 (19–26)	25 (21–31)	30 (24–39)	58 (37–78)
GammaGT U/L *	21 (15–32)	32 (22–49)	41 (26–62)	152 (70–402)
ALP U/L *	63 (53–75)	65 (56–77)	63 (51–75)	68 (65–98)
HDL mg/dL *	62 (51–75)	55 (46–67)	46 (40–53)	58 (45–70)
LDL mg/dL *	139 (116–165)	148 (124–170)	146 (118–173)	155 (121–168)
TG mg/dL *	94 (70–133)	119 (89–169)	157 (118–219)	167 (102–304)
Hb g/dL *	14.2 (13.4–15.0)	14.8 (14.1–15.6)	15.2 (14.7–16.0)	15.2 (14.3–16.5)
WBC 10⁹/L *	5.8 (4.9–6.9)	6.1 (5.2–7.2)	5.8 (5.2–6.9)	6.7 (5.5–7.3)
PLT 10⁹/L *	247 (214–283)	239 (207–273)	209 (179–241)	206 (177–219)

**Table 5 biomedicines-12-00207-t005:** Summary of rates and proportions of major and minor metabolic characteristics. The prevalence of the major characteristics is calculated for the study cohort, and the prevalence of the minor characteristics for the remaining cohort after application of the major characteristics. Within the two cohorts of major or minor criteria, one individual may be counted more than once. However, any individual is only classified once. Abbreviations: HF hyperferritinemia, FLI fatty liver index, BMI body mass index, T2DM type 2 diabetes mellitus, HOMA homeostasis model assessment, HDL high-density lipoprotein. The *p*-value indicates significant difference between the non-HF and the HF group at *p* < 0.05.

**Major Criteria within the Study Cohort n = 8408**							
		**Study Cohort** **n = 8408**	**HF All Grades n = 1111**			**Non-HF** **n = 7297**	***p*-Value**
			**HF 1** **n = 997**	**HF 2** **n = 97**	**HF 3** **n = 7**		
**Major criteria**	FLI > 60	32% n = 2669	All grades 57% (n = 636)			28% n = 2033	**<0.001**
			55% n = 544	80% n = 86	86% n = 6		
	BMI > 30 kg/m^2^	21% n = 1745	All grades 37% (n = 409)			18% n = 1336	**<0.001**
			31% n = 309	43% n = 46	57% n = 4		
	T2DM	6% n = 496	All grades 11% (n = 118)			5% n = 378	**<0.001**
			10% n = 96	18% n = 19	43% n = 3		
**Minor Criteria within the Remaining Cohort n = 5435**							
		**Remaining cohort** **n = 5435**	**Remaining HF all grades n = 437**			**Remaining non-HF** **n = 4998**	***p*-value**
			**HF 1** **n = 417**	**HF 2** **n = 20**	**HF 3** **n = 0**		
**Minor criteria**	Overweight	44% n = 2399	All grades 58% (n = 256)			43% n = 2143	**<0.001**
			57% n = 249	2% n = 7	0% n = 0		
	Arterial hypertension	44% n = 2406	All grades 53% (n = 233)			43% n = 2173	**<0.001**
			50% n = 220	3% n = 13	0% n = 0		
	Elevated HOMA-Index	14% n = 767	All grades 21% (n = 94)			12% n = 621	**<0.001**
			20% n = 88	1% n = 6	0% n = 0		
	Impaired fasting glucose	12% n = 634	All grades 23% (n = 102)			13% n = 642	**<0.001**
			22% n = 97	1% n = 5	0% n = 0		
	Low HDL cholesterol	12% n = 635	All grades 14% (n = 63)			11% n = 572	**0.067**
			13% n = 58	1% n = 5	0% n = 0		
	Elevated triglycerides	9% n = 482	All grades 12% (n = 54)			9% n = 428	**0.008**
			12% n = 51	1% n = 3	0% n = 0		

## Data Availability

The data presented in this study are available on request from the corresponding author.

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
