# Peer review of "Prevalence and Characteristics of Metabolic Hyperferritinemia in a Population-Based Central-European Cohort"

_biomedicines, 2024, doi:10.3390/biomedicines12010207_

Round 1

Reviewer 1 Report

Comments and Suggestions for Authors

The paper is well written and gives to the reader important messages about high prevalence of dysmetabolic states among subjects with HF. It has the limitation of a single time-point evaluation, like stated in the discussion. I have just minor comments, reported below. 

A limitation of this population-based study is that the cohort is poorly characterized for other causes of hyperferritinemia. For example subjects who have chronic liver disease, hepatitis, HCV infection, inflammation, or who received blood transfusions or parenteral iron, or subjects with some genetic variants may have hyperferritinemia with normal Hb and TSAT. A minimal impact of these "not so common causes" of hyperferritinemia is expected in the hyperferritinemia grade 1 and 2 groups because of high numbers of subjects, while grade 3 with only 7 subjects is very poorly informative. Even if the whole group of HF grade 3 was classified as MHF, underlining likely a role of metabolic alterations in severe HF, other causes can't be excluded. Like the Authors state in the discussion, this grade 3 may constitute an important subset for additional genetic predisposition of iron accumulation, but also additional acquired causes of hyperferritinemia should be evaluated. This message may be important for clinical practice.

I would underline in the discussion that this high prevalence of metabolic alterations in subjects with hyperferritinemia is not an established cause-effect analysis. Clinicians should be aware of the diagnostic work up in a patient with HF which may have another cause or multiple causes of HF even when they show dysmetabolic features.

Introduction, line 59: a comma is missing after variations

table 3: in the section "MINOR CRITERIA WITHIN THE REMAINING COHORT" overweight is present in 1 out of 1 subject with HF3 but % is 0

table 5: in the section "MINOR CRITERIA WITHIN THE REMAINING COHORT" overweight is present in ? subjects with HF3

Author Response

Thank you very much for your valuable comments, we hope we could answer all questions to your satisfaction.

  1. Comment: A limitation of this population-based study is that the cohort is poorly characterized for other causes of hyperferritinemia. For example subjects who have chronic liver disease, hepatitis, HCV infection, inflammation, or who received blood transfusions or parenteral iron, or subjects with some genetic variants may have hyperferritinemia with normal Hb and TSAT. A minimal impact of these "not so common causes" of hyperferritinemia is expected in the hyperferritinemia grade 1 and 2 groups because of high numbers of subjects, while grade 3 with only 7 subjects is very poorly informative. Even if the whole group of HF grade 3 was classified as MHF, underlining likely a role of metabolic alterations in severe HF, other causes can't be excluded. Like the Authors state in the discussion, this grade 3 may constitute an important subset for additional genetic predisposition of iron accumulation, but also additional acquired causes of hyperferritinemia should be evaluated. This message may be important for clinical practice.

Answer: As correctly stated, the population-based design and retrospective analysis did not allow for distinct evaluation of “less common causes” of hyperferritinemia. Nevertheless, we consider the impact of these causes in a healthy population low, although they should be evaluated during clinical workup as suggested (see changes line 261).

  1. Comment: I would underline in the discussion that this high prevalence of metabolic alterations in subjects with hyperferritinemia is not an established cause-effect analysis. Clinicians should be aware of the diagnostic work up in a patient with HF which may have another cause or multiple causes of HF even when they show dysmetabolic features.

Answer: We agree and emphasize that the high prevalence of MHF does not reflect a cause-effect relation (see changes line 265-267).

  1. Comment: Introduction, line 59: a comma is missing after variations. table 3: in the section "MINOR CRITERIA WITHIN THE REMAINING COHORT" overweight is present in 1 out of 1 subject with HF3 but % is 0. table 5: in the section "MINOR CRITERIA WITHIN THE REMAINING COHORT" overweight is present in ? subjects with HF3.

Answer: The typos in line 59, table 3 and table 5 have been corrected.

Reviewer 2 Report

Comments and Suggestions for Authors

Manuscript ID: biomedicines-2796322

Title: "Prevalence and characteristics of metabolic hyperferritinemia in a population-based Central-European cohort"

Authors: Sophie Gensluckner et al.

The authors in the present study investigated the prevalence and characteristics of Metabolic Hyperferritinemia (MHF) using the newly proposed MHF definition in a population-based Central-European cohort, which included over 8,000 individuals aged 40 to 77 years. The authors stratified participants with HF into different categories based on their ferritin levels and assessed them for metabolic comorbidities according to the MHF criteria. According to the study findings, a notable incidence of HF was observed, with a male predominance, and a large proportion of these cases met the metabolic criteria for MHF. The research highlights that HF is frequently observed in middle-aged individuals, particularly males, with the majority of them meeting the recent criteria for MHF.

The following comments should be considered:

Comments:

  1. Considering the recent consensus statement regarding the MHF diagnostic criteria, the authors should consider whether it would be more appropriate to use the newly proposed MAFLD term instead of NAFLD.
  2. The authors excluded individuals with "advanced chronic kidney disease defined by an estimated glomerular filtration rate (eGFR) of < 30 ml/min," while the "Proposed updated diagnostic criteria for metabolic hyperferritinaemia" mentioned the exclusion criterion "End-stage renal disease or dialysis." In the context of end-stage renal disease, eGFR is typically very low, usually less than 15 mL/min/1.73 m². Could you please provide further clarification on this point?
  3. Since the authors have estimated the number of individuals, it would be helpful if they could perform subgroup analyses between the non-HF and all distinct HF subgroups. If a significant change is observed, the authors should explore between which specific groups a significant difference exists (please remember to adjust for multiple comparisons). Update the statistical analysis and the other manuscript sections as appropriate. If the number of individuals in the "HF 3" subgroup does not allow for subgroup analyses, the authors could consider examining the "HF 2" and "HF 3" groups together.

Author Response

Thank you very much for your valuable comments, we hope we could answer all questions to your satisfaction.

  1. Comment: Considering the recent consensus statement regarding the MHF diagnostic criteria, the authors should consider whether it would be more appropriate to use the newly proposed MAFLD term instead of NAFLD.

Answer: According to your suggestion and the most recent definition, we changed the term from NAFLD to MASLD.

  1. Comment: The authors excluded individuals with "advanced chronic kidney disease defined by an estimated glomerular filtration rate (eGFR) of < 30 ml/min," while the "Proposed updated diagnostic criteria for metabolic hyperferritinaemia" mentioned the exclusion criterion "End-stage renal disease or dialysis." In the context of end-stage renal disease, eGFR is typically very low, usually less than 15 mL/min/1.73 m². Could you please provide further clarification on this point?

Answer: In our cohort, only one person (n = 1) had an eGFR < 15 ml/min, therefore we chose the higher cut-off of an eGFR <30 ml/min (n = 10) also according to a more practical clinical approach. We suggest that due to the low number of subjects, this does not influence the results. These numbers have been included in the manuscript.

  1. Comment: Since the authors have estimated the number of individuals, it would be helpful if they could perform subgroup analyses between the non-HF and all distinct HF subgroups. If a significant change is observed, the authors should explore between which specific groups a significant difference exists (please remember to adjust for multiple comparisons). Update the statistical analysis and the other manuscript sections as appropriate. If the number of individuals in the "HF 3" subgroup does not allow for subgroup analyses, the authors could consider examining the "HF 2" and "HF 3" groups together.

Answer: We appreciate the reviewer's suggestion regarding the performance of subgroup analyses between non-hyperferritinemia and HF subgroups (HF 1, HF 2 and HF 3). In response to this valuable feedback, we have now included two additional tables in the supplementary material of our manuscript (S4 and S5). These tables comprehensively display all p-values for comparisons between the non-HF group and each of the HF subgroups (HF 1 and HF 2 + HF 3).
We wish to emphasize that our analysis, presented in these supplementary tables, is primarily thesis-generating and descriptive in nature. Consequently, we have opted not to perform corrections for multiple testing. This decision stems from our understanding that the table describing baseline variables was not included in the formulation of any specific (null) hypotheses in the present manuscript. We believe that this approach aligns with the overall objective and scope of our study.
Moreover, we have included these considerations in the limitations section of our paper (see line 318 ff), acknowledging the potential implications of our methodological choices. We argue that presenting unadjusted p-values is more suitable for a descriptive analysis such as ours. This is because both type I (false positive) and type II (false negative) errors are plausible outcomes in our context. Therefore, we maintain that unadjusted p-values offer a more transparent and appropriate representation of our data, especially given the exploratory and descriptive nature of our analysis.
In summary, while we recognize the importance of rigorous statistical testing in many research contexts, we believe that our approach of presenting unadjusted p-values in the supplementary tables is the most appropriate method for the type of descriptive and hypothesis-generating analysis conducted in our study. We trust that this addition, along with the discussion of its limitations, will enhance the understanding and interpretation of our findings among our readers.

Reviewer 3 Report

Comments and Suggestions for Authors

This interesting study analyzes retrospectively the data of the P 10 K epidemiological  study in the area of Salzburg in Austria with the new diagnostic criteria proposed by Valenti et al for metabolic hyperferritinemia showing its high frequency .

Minor comments :

1) Did the authors have analyzed a small subset of their patients with quantitative MRI  for both liver iron and fat fraction : if so, these results could be presented and will strengthen their epidemiological findings

2) Authors should discuss the transposability of their findings to other part of the world : other part of Europa , North America even in emerging countries where there is a pandemic of overweight, obesity and diabetes mellitus.

3)In the discussion section authors discuss the relationship between iron and fatty liver disease owing to a recently published Mendelian randomization study; their discussion should also include an article demonstrating the mechanistic influence of IV iron sucrose on the generation of liver fat fraction assessed by MRI in dialysis patients (ROSTOKER G, LORIDON C, GRINUCELLI M et al. Liver iron load influences hepatic fat fraction in end-stage renal disease patients on dialysis: a proof of concept study. EBioMedicine. 2019 Jan;39:461-471. doi: 10.1016/j.ebiom.2018.11.020.) 

Comments on the Quality of English Language

English language is globally correct but can be improved by a native translator

Author Response

Thank you for your valuable comment, we hope we could answer all questions to your satisfaction.

  1. Comment: Did the authors have analyzed a small subset of their patients with quantitative MRI  for both liver iron and fat fraction : if so, these results could be presented and will strengthen their epidemiological findings.

Answer: A quantitative MRI for liver iron assessment was not part of the initial evaluation for the P10k-study and we did not perform any so far as an evaluation via MRI is not part of the consensus statement for evaluation of MHF. Nevertheless, we agree that further analysis of hepatic iron especially for the groups with moderate and severe HF could provide interesting results.

  1. Comment: Authors should discuss the transposability of their findings to other part of the world : other part of Europa , North America even in emerging countries where there is a pandemic of overweight, obesity and diabetes mellitus.

Answer: We tried to put our results in a worldwide context (see line 268f “In our cohort, the most common major criterion was fatty liver disease with an overall prevalence of 32%, which is in line with known prevalence worldwide.” or line 278-279 “The numbers for the major criteria obesity and T2DM also are in line with numbers reported previously for Europe”.) but there is, to our knowledge, no comparable data on the prevalence of MHF in other parts of the world so far.

  1. Comment: In the discussion section authors discuss the relationship between iron and fatty liver disease owing to a recently published Mendelian randomization study; their discussion should also include an article demonstrating the mechanistic influence of IV iron sucrose on the generation of liver fat fraction assessed by MRI in dialysis patients (ROSTOKER G, LORIDON C, GRINUCELLI M et al. Liver iron load influences hepatic fat fraction in end-stage renal disease patients on dialysis: a proof of concept study. EBioMedicine. 2019 Jan;39:461-471. doi: 10.1016/j.ebiom.2018.11.020.)

Answer: We appreciate mentioning this interesting work and have added it to our discussion (see line 275f)

Round 2

Reviewer 2 Report

Comments and Suggestions for Authors

Manuscript ID: biomedicines-2796322 (Revised version)

Title: "Prevalence and characteristics of metabolic hyperferritinemia in a population-based Central-European cohort"

Authors: Sophie Gensluckner et al.

The authors have tried to address my comments and suggestions, and the revised manuscript has been further improved. There are no further comments.